# Investigation to mitigate system-level factors contributing to hospital-acquired infection cases in the low-resource setting: A qualitative exploratory study in Bangladesh

Afsana Sultana [1]*, Fatema Tuz Johora [1], Palash Chandra Banik [2,3]

1 Department of Microbiology, Bangladesh University of Health Sciences, Dhaka, Bangladesh,
2 Department of Noncommunicable Diseases, Bangladesh University of Health Sciences, Dhaka, Bangladesh, 3 Centre for Higher Studies and Research, Bangladesh University of Professionals, Dhaka, Bangladesh

* afsana.t.sultana@gmail.com

## Abstract

In Bangladesh, a low-resource setting country, the range of hospital-acquired infections (HAIs) is as high as 30% in some hospitals. Several studies have been conducted to determine the prevalence of HAIs and gaps in Infection Prevention and Control (IPC) measures. However, understanding the underlying causes and the barriers in the healthcare system that prevent the implementation of IPC measures and the control of HAIs has been the least explored. The study's primary aim was to comprehend the system-level factors influencing the high prevalence of hospital-acquired infections and the barriers to implementing strict, effective IPC measures in hospitals in Bangladesh. In this qualitative study, grounded theory with an inductive approach was adopted to understand the perspective of healthcare workers and key stakeholders regarding HAIs and IPC measures. Using a non-random, purposive sampling technique, we recruited participants from four major tertiary-care hospitals in Bangladesh. 10 KIIs (Directors, Deputy Directors, Nurse Supervisors and Superintendents) and one focus group discussion (six nurses). A semi-structured interview was conducted face-to-face with the participants. To ensure data reliability, methodological triangulation was conducted. For data analysis, the constant comparative method was adopted, which involved conceptualising and categorising through open coding. The study identified multiple factors, such as gaps in IPC knowledge and practice, a lack of HAI surveillance, resource constraints, hospital infrastructure, and overcrowding, as key barriers in implementing IPC measures effectively. To prevent HAIs, besides healthcare workers and hospital management, community awareness and adherence to IPC measures also play a key role. The findings show that, to prevent HAIs, besides the healthcare workers and hospital management, community awareness also plays a key role in implementing IPC measures successfully and

**Data availability statement:** All relevant analyzed data are included within the paper and its Supporting Information files. Signed consent forms, audio recordings, and full coded transcripts are not publicly available due to ethical and confidentiality considerations. Access to these data may be granted upon reasonable request to the corresponding author and the ethics committee, subject to approval by the participants. Data Access Contact: Ethical Review Committee of the Bangladesh University of Health Sciences Email: erc@buhs.ac.bd.

**Funding:** Bangladesh Medical Research Council (BMRC), ref: BMRC/Revenue/Research Grant/2025/190(1-93). The funders had no role in the study design, data collection and analysis, decision to publish, or preparation of the manuscript.

**Competing interests:** The authors have declared that no competing interests exist.

preventing HAIs. This is why it is pertinent to develop IPC strategies that will engage both healthcare workers and the general population.

## Introduction

HAIs have been linked to several adverse outcomes, including extended hospitalization, persistent disability, increased antimicrobial resistance, amplified socio-economic disruption, and elevated mortality rates [1,2]. Hospital-acquired infections (HAIs) are more common in low and middle-income countries due to their lack of adherence to IPC guidelines [3]. The adoption of infection prevention strategies is critical for reducing the risk and occurrence of HAIs among the population, particularly in resource-constrained health systems [4].

In Bangladesh, the range of HAIs is as high as 30% in some hospitals [5]. A study conducted in a tertiary care hospital in Bangladesh showed that the prevalence of HAI in the obstetric ward, medicine ward, and surgery ward was 4.2%, 3.9% and 7.7%, respectively. Among the total post-operative patients, a high rate of 41.7% developed a surgical site infection [6].

The recent COVID-19 pandemic (Coronavirus disease 2019) revealed the gaps in the current IPC measures adopted by the Bangladesh government and tertiary care hospitals. According to the report, 44% of initial severe acute respiratory syndrome coronavirus 2 (SARS- Cov- 2) infections were acquired from hospitals. The lack of IPC measures and equipment also caused the infection of about 9,400 healthcare workers who were on the frontline from March 2020 to December 2021; among them, 180 infections resulted in deaths [7]. This emphasises why preventing HAIs by maintaining IPC measures is crucial for the security of public health and the country's economy, as HAIs increase healthcare costs, and place additional strain on already limited health system resources.

Several studies have been conducted to determine the prevalence and influencing factors of HAIS, as well as the gaps in IPC measures. However, understanding the underlying causes and the barriers in the healthcare system that prevent the implementation of IPC measures, and the control of HAIs has been the least explored. To ensure a safe hospital environment, it is essential to identify the root causes of the failures in implementing IPC measures and reduce the rate of HAIs in hospitals in Bangladesh. The study's primary objective was to comprehend the system-level factors influencing the high prevalence of hospital-acquired infections and the barriers to implementing strict, effective IPC measures in hospitals in Bangladesh.

## Methods

### Study design and theory

For this study, grounded theory with an inductive approach was adopted to understand the perspective of healthcare workers and key stakeholders regarding HAIs and IPC measures. This approach was well-suited for examining system-level factors, as it focuses on understanding processes, interactions, and organizational

dynamics and allows theory to emerge from healthcare service providers' experiences, particularly where established models do not fully capture contextual and institutional influences. A semi-structured guideline was developed to understand the current situation of HAIs; strategies implemented to prevent HAIs and maintain IPC measures in the hospitals. Using this method allowed us to understand the experiences and perceptions about HAIs and IPC implementation in detail. Furthermore, in this study, the grounded theory guided iterative analysis using the constant comparative method, with open, axial, and selective coding applied to identify relationships between organizational processes, resource constraints, and staff practices.

## Settings and participants

A non-random, purposive sampling technique was adopted to recruit participants from four tertiary level hospitals in Bangladesh between November 2024 to January 2025. The study was conducted in Dhaka Medical College Hospital, Shaheed Suhrawardy Medical College and Hospital, a 250-bedded TB Hospital and the Infectious Diseases Hospital, which are considered major hospitals for the treatment of serious communicable diseases in Bangladesh, providing health services to thousands of patients regularly. KII was conducted until the data was saturated. After the eighth KII, interviewers found no new concept or codes/themes, which was confirmed during the last 2 KIIs and 1 FGD. As a result, the study conducted 10 KIIs and 1 FGD, which included 6 nurses.

10 Key informant interviews (KII) were conducted with the nurse supervisors, superintendents, hospital Director, Deputy Director and a nurse with experience in HAI management of four tertiary care hospitals. A focus group discussion (FGD) with nurses was conducted in a department with a high prevalence of HAI in Dhaka Medical College Hospital (DMCH). For KIIS and FGDs, appointments were taken in advance, and the interviews were held at their preferred time in their workplace. The KIIs and FGD were conducted by the first two authors, both the female authors held bachelor's-level degrees and were employed as research assistants. The interviewers had prior exposure to qualitative research methods and underwent training, including a pilot (trial) interview with intern nurses, to refine the interview guide and ensure quality in data collection. To ensure transparency and comprehensiveness of the interviews a COREQ checklist is provided in S1 File. Study process

## Data collection process

A semi-structured interview was conducted for KIIs, and FGD. 10 KIIs and 1 focus group discussion were conducted face-to-face. Ethical permission was taken from the Ethical Review Committee, Bangladesh University of Health Sciences (BUHS/ERC/EA/24/60). Before the interview, participants were briefed about the objective of the research and what was expected of them during the interview. The informed consent included the risks and benefits of participating in the study. Participants had the right to withdraw at any time during the interview or could opt not to answer some questions and could seek anonymity. Consent was taken explicitly to audiotape their interview. Both written and verbal consent were taken before and after starting recording. To protect the confidentiality of the healthcare workers in this study, their names and department names have been omitted. The interviews were recorded with a tape recorder with the participants' consent.

For KII and FGD, the interviews were conducted following the standard procedure. Interviews took place in quiet private rooms (for KII) and a quiet area (for FGD) with the participant and research team. No prior relationship existed between the interviewers and participants before study commencement. At the time of recruitment and before the interviews, participants were informed that the interviewers were members of the research team and that the objectives of the study. As the study employed grounded theory, no prior biases or assumptions by the interviewers were identified. Each KIIs lasted for 10–25 minutes, with an average of 17 minutes, and the focus group lasted for 21 minutes. The questions in the guidelines were regarding the prevalence of HAIs, the current IPC protocol, safety measurements for healthcare workers, challenges in maintaining infections in their respective hospitals and suggestions for improving HAI prevention measures. The data collection materials are provided in the S2 File. The sample size depended on theoretical saturation being achieved.

## Data analysis

All the interviews were initially transcribed into the local language (Bengali) and after that translated into English by a professional. To ensure consistency of the transcription, the transcribed data in English was back-translated by another professional. For data analysis, the constant comparative method was adopted, which included conceptualising and categorising through open coding, followed by axial coding, and, finally, the phenomena were defined through selective coding [8]. The code was used to define and mark the experience, feelings, and suggestions of the participants, but not the hearsay accounts.

As data was collected through FGD and KIIs, methodological triangulation was conducted. Two investigators independently reviewed the primary codes to form axial coding and then compared them to identify the variances. Disagreements that were not resolved through consensus between the two investigators were resolved by the third investigator. After resolving the variance, the codes were categorised for thematic analysis. Coding was done by highlighting sections of the transcribed text, usually phrases or sentences, and short labels were used to describe the codes. Each code describes the idea or feeling expressed in that part of the text. The codes were used to identify patterns among them and generate categories and, eventually, themes. For data management and analysis, the software Taguette was used [9]. Table 1 shows the codes, categories and themes that were used in the study. Additionally, an excerpt from the codebook is provided in the S3 File.

# Results

## Participants characteristics

In the KIIs, the mean ± SD of age and work experience in their role were 49.6 ± 10.4 and 25.3 ± 11.8, respectively (Table 2). In the FGD, the mean ± SD of age and experience of the nurses were 41.3 ± 10.1 and 17.3 ± 8.4, respectively (Table 3).

## Themes

A total of 10 themes were identified. Most participants were able to answer with a few examples of their experiences and feelings. They were able to identify the shortcomings and barriers and provide suggestions based on their experiences working in a hospital setting. As a result, we were able to identify various patterns, which are summarised in the following themes.

**Theme 1: Need for systematic HAI monitoring.** When asked about the prevalence of hospital-acquired infections in their respective hospitals, most participants admitted the presence of HAIs in their hospital but were unable to provide reliable or specific data, which they attributed to the absence of routine surveillance, standardized data audits, or regular prevalence studies at the institutional level

*"At least in Surgical wards, having no nosocomial infection is very rare, but an incidence or prevalence study in a whole hospital has not been conducted."* (KII2)

According to the Nurse Supervisor of DMCH, they can understand just through observation. Most of the participants from the four hospitals reported that they do not conduct regular surveys or monitoring. However, they occasionally conduct laboratory tests or surveys when they get overwhelming reports from specific departments to identify the causes and minimize the infection. According to the nurse informant, the reason for the lack of statistical data for HAIs was the lack of a monitoring system and the overwhelming patients, making it difficult for them to determine whether the infection was acquired from the hospital or outside.

**Theme 2: Overburdened healthcare workforce.** Maintaining IPC and preventing HAI is challenging for healthcare workers as they are overburdened due to the significant difference in the nurse-patient ratio, which also puts their health

**Table 1. Identified themes of the hospital-acquired infection study conducted among health professionals in various tertiary care hospitals in Dhaka city, 2024.**

| Code | Category | Theme | Quotes |
|---|---|---|---|
| Lack of Baseline Data | Surveillance and Research | Need for Systematic HAI Monitoring | *"Nosocomial infections do occur but cannot be accurately detected in our hospital due to a lack of proper monitoring systems."* (KII10) |
| Nurse-Patient Ratio | Staffing Issues | Overburdened Healthcare Workforce | *"Nurses often work with a 1:99 ratio, making it impossible for one person to handle 99 patients."* (KII10) |
| Lack of IPC Training | Infection Prevention and Control | Gaps in IPC Knowledge and Practice | *"Yes, they (trainers) come sometimes, sometimes we see the training is off. Some receive training, some don't. I didn't get the training."* (FGD3) |
| Overcrowding<br><br>Environmental Factors | Hospital Infrastructure | Impact of Hospital infrastructure and overload | *"There is a weakness in the infrastructure. In 1972, these 200-bedded hospitals became 2,600-bedded hospitals. So, when an infrastructure of a 200-bedded hospital is converted into a 2600-bedded hospital, definitely people would face problems"* (KII2) |
| Poor Waste Management | Cross-Contamination and Hospital Hygiene | Hospital Hygiene | *"There are no specific trainings or guidelines given on waste management, and no one follows them."* (KII10) |
| Lack of PPE and Supplies<br><br>Syringe and Equipment Reuse | Resource Shortages and Risk Exposure<br><br>Unsafe Medical Practices | Consumables and Resource Constraints | *"Everyone knows about the shortage. Our ward-in-charge requests supplies, but they are often not fulfilled due to lack of availability"* (FGD2) |
| Nosocomial Infection Patterns | Infection Epidemiology | Common HAIs and At-Risk Groups | *"No surgery ward can say there aren't 1 or 2 cases of infection after the operation".* (KII2) |
| Emotional and Physical Burden on Nurses | Workplace Safety | Healthcare Worker Safety Concerns | *"Many of our colleagues faced problems. Many have been at life risk, suffering from respiratory issues and so on".* (KII3) |
| Overcrowding<br><br>Healthcare Worker Negligence<br><br>Community Awareness<br><br>Limitations | Hospital Infrastructure<br><br>Professional Practices and Public Engagement | Factors influencing the rate of HAIs | *"Everyone is somewhat responsible. The patient is throwing garbage outside, the cleanliness sweeper isn't cleaning, and I am not monitoring properly. So, on an individual level, everyone from patients to doctors, nurses, and staff is somewhat responsible. On a family level, they have attitude problems; they should be given a lesson, to not visit more than one attendant. For 1 patient, 10 attendants come, then there is a factor too."* (KII2) |
| Policy-Level Strategy<br><br>Facility and Community Level Strategies<br><br>Implemented strategies | Policy-Level Strategy<br><br>Facility and Community Level Strategies | Strategies for Reducing HAIs | *"This is an integrated approach; you and we should adopt a holistic approach, and only then will the IPC be effective."* (KII2) |

Data analysis progressed from inductive coding to category development and theme generation, with constant comparison across transcripts.

at risk. Key informant interviews from nurse supervisors, nurse superintendents, and focus group discussions significantly highlight this issue. It is pertinent to mention that the DMCH is the largest tertiary care hospital in Bangladesh, which serves about 4160 patients regularly. According to the Deputy Director of DMCH, there are 2600 beds in the hospital with only 600 hospital staff. The Director of DMCH also added that, currently, on average, each hospital bed has 1.6 patients, as in many instances, patients have to share a bed with other patients. The nurse superintendent of DMCH stated, they have to take care of 3 patients instead of 1. As a result of an overburdened workforce, nurses can face difficulties in maintaining IPC guidelines and providing high-quality care to their patients.

**Theme 3: Gaps in IPC knowledge and practice.** This study has found gaps regarding IPC knowledge among the healthcare workers, resulting in them having poor IPC practice. While most of the hospitals in our study had an IPC committee, the committee lacked effectiveness in preventing HAI, as there was no regular IPC training for healthcare workers or monitoring to ensure the IPC measures were maintained. According to the participants, the training isn't

**Table 2. Characteristics of the KIIs participants of the hospital-acquired infection study conducted among health professionals in various tertiary care hospitals in Dhaka city, 2024.**

| Participants | Institution | Position | Age (yrs) | Experience (months) | Total work experience (yrs) |
|---|---|---|---|---|---|
| KII1 | Dhaka Medical College Hospital (DMCH) | Director | -- | 19 | -- |
| KII2 | Dhaka Medical College Hospital (DMCH) | Deputy Director | 52 | 7 | 26 |
| KII3 | Dhaka Medical College Hospital (DMCH) | Nurse supervisor | 56 | 120 | 30 |
| KII4 | Dhaka Medical College Hospital (DMCH) | Nurse superintendent | 59 | 10 | 42 |
| KII5 | Shaheed Suhrawardy Medical College and Hospital (SSMCH) | Director | 54 | 16 | 27 |
| KII6 | Shaheed Suhrawardy Medical College and Hospital (SSMCH) | Deputy Director | 56 | 24 | 30 |
| KII7 | 250-bedded TB Hospital | Residential Doctor | 38 | 8 | 10 |
| KII8 | 250-bedded TB Hospital | Nurse Superintendent | 57 | 54 | 34 |
| KII9 | Infectious Diseases Hospital | Hospital super/ Senior Consultant (medicine) | 46 | 60 | 22 |
| KII10 | A tertiary care hospital | Nurse | 28 | 36 | 7 |

Age and total work (professional) experience are reported in years. Experience (months) refers to the duration of participants' service in their current position or department. "--" indicates information not disclosed by the participant.

**Table 3. Characteristics of the FDG participants of the hospital-acquired infection study conducted among health professionals in various tertiary care hospitals in Dhaka city, 2024.**

| Participants | Gender | Age (yrs) | Position in the ward | Work experience (yrs) |
|---|---|---|---|---|
| FDG1 | Female | 50 | Senior staff nurse | 25 |
| FGD2 | | 35 | | 10 |
| FGD3 | | 29 | | 9 |
| FGD4 | | 33 | | 10 |
| FGD5 | | 48 | | 24 |
| FGD6 | | 53 | | 26 |

**Age and total work experience are reported in years. Work experience (refers to the duration of participants' service as a nurse.**

regular and is usually reserved for nurses in senior positions. The restriction of IPC training to senior nursing staff highlights both deficiencies in cascade training mechanisms and a lack of sustainable training infrastructure. While senior staff may receive IPC instruction, the absence of formal knowledge-transfer systems limits dissemination to junior and support staff who also play an important role in HAI prevention. Additionally, irregular training schedules point to the lack of a structured, system-wide IPC training programme.

When asked about the IPC committee and training, the Directors, Deputy Directors, and Hospital Super/Consultant stated that they have a strong IPC committee. However, the DMCH nurse supervisor, the nurse superintendent from KII, and nurses from the focus group discussion had different opinions.

*"Yes, they (trainers) come sometimes, sometimes we see the training is off. Some receive training, some don't. I didn't get the training"*. *"Neither did I"*, nurse no. 4 added. Nurse no. 2 stated that, *"When we ask the nursing supervisor, they say the training has been discontinued due to lack of funding."* (FGD3)

**Theme 4: Impact of hospital infrastructure and overload.** While environmental factors were an important factor in preventing HAI and maintaining IPC measures, overload due to overcrowding was highlighted the most as one of the main causes during KIIs and FGD. The participants believe overcrowding is a limitation in maintaining IPC measures.

*"During the daytime, the population in Dhaka Medical is, can you imagine, 40,000 people come and go in this building. Each of my 4300 patients has at least 2/3 attendants. In the outdoor 5000 patients come, and each of them brings 2/3 attendees, and then indoors, a total of 30,000 here and my staff is 6000, Ansar (a paramilitary auxiliary force in Bangladesh), reporters, and hawkers, in total 40,000 people. Now that 40,000 people have walked here and spread dust, with my manpower cleaning this, how many (cleaning) teams are prepared?"* (KII2)

According to the participants, the main cause of this issue is the infrastructure, as, according to them, even though the number of patients and staff increased, the facility has not grown in size. The Deputy Director of DMCH provided an insight into the infrastructure of the largest tertiary care hospital in Bangladesh.

*"The hospital is a 130-year-old building, and this is not the hospital structure; this was a governing body's office of Assam state. During the 2nd World War, it was converted into a 200-bed hospital. But it's not a hospital structure; the height and ventilation of the building were not for a hospital. The architecture of hospitals is different since the amount of sunlight, air and lighting in the hospital is specified. There is a weakness in the infrastructure. In 1972, these 200-bedded hospitals became 2,600-bedded hospitals. So, when an infrastructure of a 200-bedded hospital is converted into a 2600-bedded hospital, definitely people would face problems"* (KII2)

**Theme 5: Hospital hygiene.** The study found that a lack of proper waste management is responsible for cross-contamination, resulting in health risks for both the healthcare workers and patients in the hospitals. According to the participants, all the infectious waste, without any classification, mostly goes to yellow colored bin, and the sharp object goes to the red-colored bin.

*"For the waste bins, there are 3 criteria. One for normal waste, another for liquid waste, and one for sharp and cutting waste."* (KII5)

According to the participants, even if they use separate bins, the workers mix everything into one single bin and usually do not even wear gloves when handling these infectious materials. Same participants mentioned, even though they try to maintain waste management, they lack the manpower and logistics to fully meet the standards, and also, due to the patients and their relatives, they can't maintain the waste management that much.

Meanwhile, in the 250-bedded TB hospital, according to the Resident Doctor, most of the protocol is maintained. To prevent patients from coughing up sputum anywhere, they have placed bins in different places. He believes that since all the patients who come to the hospital are aware of the contagiousness of the diseases that are treated in the TB specialised hospital, they easily comply with the safety guidelines.

**Theme 6: Consumables and resource constraints.** Lack of safety and medical equipment can cause healthcare workers to reuse the equipment, which may not always be sanitary and sterilised enough to be used, resulting in unsafe medical practice. Most healthcare workers, including nurse supervisors, nurse superintendents, and senior staff nurses, have reported a shortage in healthcare and safety equipment, resulting in a risk to the health of both healthcare workers and patients.

Nurse 4 stated, *"Operating rooms have so many patients, compared to the patients, the equipment is less, and we are forced to reuse the same instruments multiple times after sterilization. No matter how much we autoclave or maintain sterilisation, it's never enough, it will never be done properly"* (FGD4).

According to some participants still a practice of reusing catheters and syringes for the same patients by washing them with only alcohol.

*"Most importantly, the issue lies with syringes. Ideally, a single syringe should be used for one patient and discarded. But many times, discarding syringes isn't possible—for instance, over the last six months, we haven't had a sufficient supply of syringes. Some patients who can afford it manage to buy their syringes, but for others, we are compelled to reuse the same syringes. This is a significant problem"*. (KII10)

When asked why they do not inform the authorities regarding the shortage, Nurse 2 stated that, *"Everyone knows about the shortage. Our ward-in-charge requests supplies, but they are often not fulfilled due to lack of availability"* (FGD2).

**Theme 7: Common HAIs and at-risk groups.** When asked about the most of HAIs, participants named respiratory, reproductive tract, wound infections and blood infections. And nurses, women, children and immunocompromised people are the most at risk for HAIs. According to the participants, HAIs commonly occur in the ICU, post-operative surgical care and the gynaecology department. This results in late recovery, which significantly impacts the health outcome of the patients, and a longer stay at the hospital results in a financial burden.

*"No surgery ward can say there aren't 1 or 2 cases of infection after the operation"*. (KII2)

**Theme 8: Healthcare worker safety concerns.** During KIIs and FGD, many mentioned that healthcare workers, especially nurses, are at the most at risk of HAI due to inadequate safety equipment, protocols and as they have direct contact with the patients.

According to the participants, since patients are unaware if they are contagious or sometimes intentionally hide their contagious diseases, and because patients believe they might face discrimination or be refused treatment, nurses have to care for the patients without full information. Participants also added that patients do not adhere to the IPC rules, resulting in risking the health of both the healthcare workers and the patients.

*"When it comes to skin diseases, for instance, if one patient has a skin infection, the proximity between patients often allows the infection to spread easily. Even when we provide instructions, they often fail to maintain proper hygiene. This is a risk not just for them but also for us healthcare workers. As we work near patients, there is always a risk"*. (KII10)

**Theme 9: Factors influencing the rate of HAIs.** The study participants considered overcrowding, healthcare worker negligence, community awareness, and some limitations in the hospitals' management are influencing the rate of HAIs.

All the participants in the study have acknowledged that overcrowding and the lack of community awareness have hindered the implementation of IPC measures properly, additionally also hampering the work of healthcare workers.

*"You can visit another ward. When you enter the female ward, you wouldn't feel it's a female ward; in every bed, you will see 2/3 males with the women. This inevitably spreads infections. A post-operative patient should not be surrounded by so many people, yet their relatives stay beside them, leading to infection risks. Some C-section patients return with infections at their incision sites. This cycle continues, and we, as healthcare workers, also suffer exposure"*. (FGD2)

While it is evident that controlling overcrowding is important by limiting attendees, there are many limitations. According to the nurse informant, there is no dedicated team or proper control mechanism to control overcrowding. The Deputy

Director of DMCH also stated, the reason they can't stop the entry of attendants is that, unlike hospitals in other countries, hospitals like DMCH in Bangladesh are not self-sufficient.

> "*In the middle of OT, we (healthcare providers) allow attendants to tell them we need surgical threads. So, we have to trade off, and balance, how many attendants I will allow*". (KII2)

The Deputy Director of DMCH believes that the lack of awareness regarding IPC and infection among patients, doctors, nurses, and staff is somewhat responsible for the spread of infections. The Deputy Director of SSMCH identified the lack of proper sterilisation and proper OT facilities, and overcrowding as the causes.

Besides this, during KIIs, many have pointed out the negligence of healthcare workers as one of the main causes influencing the prevalence of HAIs in hospitals. According to the Deputy Director of DMCH, they provide the support, and the nurses know of it; however, they lack a positive attitude and practice towards maintaining IPC regulation. He believes the nurses need to develop self-esteem, be motivated, and sincere, otherwise the training and monitoring wouldn't be effective. When asked why IPC isn't maintained, the nurse supervisor replied, "*You could say negligence.*"

**Theme 10: Strategies for Reducing HAIs.** According to the participants, strict government health policies, IPC training, improved infrastructure, adequate medical supplies, and community awareness are important to reduce the rate of HAIs in the hospitals in Bangladesh.

> "*If we just create strategies on pen and paper like many other current strategies, such as child marriage, smoking. So, we shouldn't only be dependent on strategy (on paper) but also need to ensure it's implemented. Another thing is that our every strategy, including maximum health strategies, does not include community engagement".* (KII2)

To prevent overcrowding, the participants emphasised the importance of opening more emergency wards, which need to be positioned in a different section of the building. To prevent HAIs, the nurse superintendent of DMCH proposed maintaining proper surgery/operation guidelines and ensuring patients follow post-operation instructions. She also added that it is important to bring awareness among nurses and the public about maintaining hygiene. Similarly, the Director of SSMCH proposed that, besides creating awareness among the healthcare workers in the operating theatre and maintaining equipment sterility, the visitors' movement needs to be controlled.

According to the Residential Doctor in the TB hospital, some of the implemented initiatives to reduce the risk of HAI are providing instructions to the patients, such as wearing a mask, in the reception area. He proposed that showing health educational videos on Bangladesh's national TV and social media will be an effective way to create awareness among the general population.

## Discussions

The prevention of HAIs is key to improving the healthcare system of a country. Infection prevention and control measures by the WHO have been proven effective in reducing the rate of infection in hospitals. However, frequent HAI cases have been reported in various tertiary care hospitals in Bangladesh. This is why the study aimed to identify the factors and barriers to implementing IPC measures to prevent HAIs in hospitals. The study identified multiple factors, such as gaps in IPC knowledge and practice, a lack of HAI surveillance, hospital infrastructure, and overcrowding, as key barriers in implementing IPC measures to prevent HAIs.

For the prevention of HAIs and maintaining the safety of healthcare service providers and service recipients, IPC guidelines play a crucial role. However, the study found significant gaps in IPC knowledge and practice. The study identified factors such as lack of IPC training, PPE, and. supplies, community awareness, and healthcare workers' negligence impacting the IPC knowledge and practice of healthcare workers, patients, and visitors. Studies on the Knowledge,

attitude, and practice of nurses regarding HAIs and IPC found that poor knowledge and practice [10,11]. However, no studies were conducted to identify the IPC knowledge and practices of healthcare service recipients. Due to having inadequate knowledge about HAIs and IPC guidelines, patients and their families, and sometimes some healthcare providers, can contribute to the spread of infectious diseases among healthy individuals. For community awareness regarding IPC and HAIs, patients and their families need to be provided with counselling in the waiting area. To create community awareness on a large scale, awareness campaigns through television and social media should be done. To ensure healthcare workers are motivated to follow IPC guidelines, hospitals should create a dedicated IPC team that will provide IPC training, ensure adequate PPE, and monitor adherence to IPC guidelines and HAI rate.

According to this study's findings, currently, there is no accurate database or effective surveillance to identify the rate of hospital-acquired infections in the four major tertiary care hospitals in Bangladesh. This reflects a wider national gap in infection surveillance, as other studies also suggest that about 90% of the tertiary care hospitals in Bangladesh do not strictly monitor the IPC measures or even have an audit system, and only 30% have an IPC training program regularly [12,13]. Strengthening national surveillance and mandating routine reporting of hospital-acquired infections should therefore be prioritized within Bangladesh's health policy framework to improve patient safety and healthcare quality.

The current study found various factors contribute to the lack of surveillance for HAIs in hospitals in Bangladesh. Such as, lack of manpower, as stated by the Director of DMCH and the Hospital Super of Infectious Hospital, resulting in having no designated team for regular monitoring HAIs and IPC implementation. As a result, a high rate of infection can be observed in various hospitals' wards that should maintain strict sterility, compromising the health of both patients and healthcare workers.

Besides unmonitored HAI burden and gaps in IPC knowledge and practice, it is estimated that the cause of the high prevalence of HAIs in hospitals in Bangladesh is the poor hospital infrastructure. In Bangladesh, 55% of hospitals do not have adequate space to accommodate the high number of patients, 73% of hospitals do not have hand-wash stations, and 37% do not have an adequate number of toilets [14]. Most of these tertiary care hospitals in our study have been established during the 1940s to 1960s, with no significant recent developments. For instance, Dhaka Medical College and Hospital is one of the largest tertiary care hospitals in Bangladesh. However, its infrastructure does not comply with the standards for hospital infrastructure, as the building wasn't initially made for hospital purposes. Poor design obstructs healthcare workers from providing a healthy environment and sometimes handling emergency cases. Most of the hospitals in our study had closed-off wards, with no scope of air and sunlight from outside. Many nurses in the FGD mentioned that the wards had high temperatures and no dedicated negative-pressure room. Patients and visitors can be mostly seen cramped in closed-off wards, sometimes exceeding the capacity of the wards' space. The small number of washrooms and waste bins can hardly accommodate so many patients and their families. The poor waste management, dirty and wet washrooms, and lack of hand-washing basins with clean water add to the unhygienic environment [15,16].

The infrastructure of hospitals in Bangladesh creates a suitable environment for pathogens to grow and prevents to keep of a sterile environment for both the healthcare workers and patients. This is why it is important for the government to develop or improve the infrastructure of these hospitals with an updated design following the WHO guidelines.

Additionally, the study also identified overcrowding as the biggest obstacle in maintaining IPC measures. However, with the current capacity of hospitals, it has become challenging to control visitors to prevent overcrowding. Participants mentioned that they have to allow patients' visitors as the hospitals are not self-sufficient. Patients' families are most of the time needed to bring medicines, syringes, etc. Even during surgeries, the patients' families are required most of the time to manage blood for transfusion due to the inept blood bank system in hospitals in Bangladesh [17]. Besides the shortcomings of hospital management, overcrowding is also caused at the community level. In Bangladesh, the social bond among the people is strong, which makes it a familial duty for the family and relatives to visit the patients in the hospital [18]. And due to a

lack of community awareness regarding HAIs and IPC, these duties are usually fulfilled by compromising IPC measures, such as not wearing masks or touching the patient's bandages. Moreover, it is important to note that, in Bangladesh, the capital city, Dhaka, accommodates the most tertiary care and peripheral hospitals, consequently receiving patients from all over the country. This results in overcrowding, which the current infrastructure and the limited manpower of these hospitals are not prepared to handle. Which is why, besides improving the infrastructure of current hospitals, the government should take initiatives to decentralise Dhaka and develop more tertiary care hospitals outside Dhaka city.

Additionally, it is pertinent to ensure that, when implementing IPC guidelines, the community is also included. Our findings in the 250-bedded TB hospitals show that including the community helps significantly in implementing and maintaining IPC measures effectively in the hospital. The hospital provided counselling to the patients and the visitors from the beginning, and provided adequate trash bins for the patients to cough up their sputum. The difference in implementing IPC measurement between the 250-bedded TB hospital and the other three hospitals shows the importance of community awareness. Participants perceived that healthcare workers, patients, and visitors at the TB hospital demonstrated good adherence to IPC protocols, which they attributed to higher awareness of TB-related hazards in the community. However, as no empirical studies have yet examined the role of community awareness in influencing IPC adherence, this observation should be interpreted as a hypothesis generated from qualitative insights that warrants further investigation.

Furthermore, social and cultural factors, particularly community engagement and visitor behavior, play an important role in maintaining a safe hospital environment. To ensure cooperation between the community and healthcare service providers in creating a safe hospital environment, it is important to implement risk communication between healthcare providers and recipients [19]. Effective risk communication helps build trust among healthcare providers and recipients, ensures patient safety, and enables effective emergency response during any disease outbreak [20].

In contrast, several challenges identified in this study reflect structural health system limitations, particularly related to hospital infrastructure and patient flow management. In line with the WHO infection prevention and control (IPC) and WASH in health care facilities guidelines, the zoning method should be adopted [19,21] Colour-coded zones should be designed based on the severity of patients' conditions; visitor movement should be limited or restricted to those zones accordingly. This will help ensure the safety of both patients and visitors [22]. Furthermore, implementing the structural triage recommended by WHO for low-resource settings will also assist in addressing manpower and medical equipment shortages [20]. Triage ensures resources are allocated according to patients' needs, resulting in more effective treatment and preventing long waiting times, especially for critical patients [22].

The qualitative study allowed us to understand the influencing factors of HAIs and barriers to implementing HAIs on a systematic level in depth. The use of grounded theory was a strength as this approach allowed exploration of HAIs as a system-generated phenomenon rather than isolated behavioural issues. By focusing on processes and interactions, the approach helped identify how resource shortages and institutional practices collectively shape infection risks, supporting the development of a contextually grounded explanation of IPC challenges in low-resource hospital settings.

However, there are some limitations to the study. While the study mainly focused on the perspective of healthcare service providers, the perspective of healthcare service recipients was not taken due to a lack of baseline data on patients and the community perspective on HAIs and IPC. Additionally, the lack of quantitative data also resulted in our not being able to provide quantified data for our findings.

## Conclusions

The study addresses the gaps identified in the implementation of IPC (Infection Prevention and Control) and HAI (Hospital-Acquired Infection) prevention in previous studies. The study identified major obstacles to maintaining IPC and preventing HAIs in depth, allowing for the identification of potential solutions that will be feasible in low-resource settings. For instance, the study found that, while overcrowding is one of the major obstacles, due to some specific economic and

social factors, restricting visitors is difficult in low-resource settings. Henceforth, the findings will not only help identify the factors that need to be focused on but also how to mitigate these challenges to create an environment and a socially suitable IPC framework. The findings are important for creating government policies and guidelines to prevent HAIs and effectively implement IPC measures, even during a national health emergency in a low-resource setting.

In Bangladesh, public hospitals are affordable and serve as the primary treatment option for the majority of the population. However, due to the ongoing issue of HAIs, people risk their health to access low-cost treatment in government hospitals. This highlights the urgent need for action to address this serious concern. The findings indicate that preventing HAIs requires not only the efforts of healthcare workers and hospital management but also increased community awareness and strict adherence to IPC measures. Developing IPC strategies that involve both healthcare workers and the general population is therefore essential. To improve compliance with IPC rules among healthcare providers and patients, raising awareness within the community and among healthcare workers is vital. Regular training and sufficient medical and safety resources for healthcare staff can help close gaps in their knowledge and practice of HAI prevention and IPC protocols. Concerning hospital infrastructure, adopting the zoning method by implementing colour-coded zones based on patients' severity is recommended. Visitor movement should be restricted or limited to appropriate zones to ensure the safety of both patients and visitors. Additionally, implementing triage can help address shortages of manpower and medical equipment.

## Supporting information

**S1 File. COREQ checklist.**
(DOCX)

**S2 File. Data collection materials and additional quotes.**
(DOCX)

**S3 File. Codebook.**
(XLSX)

## Acknowledgments

The authors thank Dhaka Medical College Hospital, Shaheed Suhrawardy Medical College and Hospital, the 250-bedded TB Hospital and the Infectious Diseases Hospital for their cooperation in this study.

## Author contributions

**Conceptualization:** Afsana Sultana, Fatema Tuz Johora, Palash Chandra Banik.

**Data curation:** Afsana Sultana, Fatema Tuz Johora.

**Formal analysis:** Afsana Sultana, Fatema Tuz Johora.

**Funding acquisition:** Palash Chandra Banik.

**Investigation:** Afsana Sultana, Fatema Tuz Johora, Palash Chandra Banik.

**Methodology:** Afsana Sultana, Fatema Tuz Johora, Palash Chandra Banik.

**Supervision:** Palash Chandra Banik.

**Visualization:** Afsana Sultana, Fatema Tuz Johora, Palash Chandra Banik.

**Writing – original draft:** Afsana Sultana.

**Writing – review & editing:** Afsana Sultana, Fatema Tuz Johora, Palash Chandra Banik.

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
