## [Decision Letter · Decision Letter 0]

18 Dec 2025

Dear Dr. Sultana,

 The manuscript has been evaluated by two reviewers, and their comments are available below. The reviewers have raised a number of concerns with the methodology in your manuscript that need attention. Could you please revise the manuscript to carefully address the concerns raised?

We look forward to receiving your revised manuscript.

Kind regards,

Brian Patrick Weaver, Ph.D.

Staff Editor

PLOS One

Journal Requirements:

“This study was funded by the Bangladesh Medical Research Council (BMRC), ref: BMRC/Revenue/Research Grant/2025/190(1-93). The authors thank Dhaka Medical College Hospital, Shaheed Suhrawardy Medical College and Hospital, the 250-bedded TB Hospital and the Infectious Diseases Hospital for their cooperation in this study.”

“Bangladesh Medical Research Council (BMRC), ref: BMRC/Revenue/Research Grant/2025/190(1-93)”

“Bangladesh Medical Research Council (BMRC), ref: BMRC/Revenue/Research Grant/2025/190(1-93)”

Reviewers' comments:

Reviewer's Responses to Questions

**Comments to the Author**

1. Is the manuscript technically sound, and do the data support the conclusions?

Reviewer #1: Yes

Reviewer #2: Yes

2. Has the statistical analysis been performed appropriately and rigorously?

Reviewer #1: Yes

Reviewer #2: Yes

3. Have the authors made all data underlying the findings in their manuscript fully available?

Reviewer #1: Yes

Reviewer #2: Yes

4. Is the manuscript presented in an intelligible fashion and written in standard English?

Reviewer #1: Yes

Reviewer #2: Yes

Reviewer #1: Thank you for the opportunity to review this important and timely manuscript. The authors present a qualitative exploratory study investigating system-level barriers to infection prevention and control (IPC) practices and the high prevalence of hospital-acquired infections (HAIs) in low-resource tertiary hospital settings in Bangladesh. The manuscript provides valuable insights that could contribute meaningfully to policy and practice in infection control within similar contexts.

The study is well-conceived, with a clear research aim and relevance. The inclusion of both key informant interviews and a focus group discussion offers a multi-perspective view on structural and institutional challenges. However, there are some methodological and reporting areas that would benefit from strengthening. My detailed feedback is provided below, structured according to key domains of qualitative research-

1. Study Design and Reporting (COREQ Adherence)

The paper presents an inductive grounded-theory approach and semi-structured guides, with purposive sampling across four tertiary sites. While this is appropriate and clearly articulated, core elements of qualitative reporting per the COREQ (Consolidated Criteria for Reporting Qualitative Research) checklist are currently underdeveloped. Examples include-

* Researcher characteristics and potential influence on data collection.

* Nature of the relationship (if any) between interviewers and participants.

* Setting of interviews (e.g., location, privacy conditions).

* Whether the interview guide was pilot-tested or iteratively refined.

Since the adherence to COREQ is a standard practice for studies of this nature, I strongly recommend the authors to include a COREQ checklist as a supplementary file and revise the Methods section to align with its domains.

2. Sample Size and Saturation

The study includes 10 key informant interviews (KIIs) and 1 focus group discussion (FGD) with 6 nurses. The authors state that data collection continued until saturation was reached. Please clarify how “theoretical saturation” was operationally defined (e.g., no new codes/themes emerging across the last few interviews).

3. Data Collection Procedures

The manuscript reports face-to-face interviews, their durations, and describes consent procedures well. However, a few critical methodological details require clarification:

* Who conducted the interviews? What training did they have in qualitative methods?

* Were the interviews held in private settings to mitigate social desirability bias?

* Did the interview guides evolve as data collection progressed (i.e., iterative refinement)?

Such clarifications will enhance the credibility of the findings.

4. Data Analysis and Tools

The use of constant comparative analysis is appropriate and well-described. However, there are a few inconsistencies and opportunities for improvement:

* The software cited, “Tageutte,” appears to be a misspelling. It is likely “Taguette.” Please confirm and correct this throughout the manuscript, and provide version details and URL (as is customary for citing statistical software)

* Including an excerpt from the codebook (as supplementary material) would strengthen the transparency of the analytical process.

5. Ethics and Consent

Explain why both written and verbal consent were obtained. For example, was verbal consent needed to confirm agreement to audio-recording? Also, indicate whether consent forms are available on request, in line with PLOS data availability policies.

6. Interpretation and Policy Implications

The manuscript notes that staff at the TB hospital “showed greater adherence to the IPC protocol,” attributing this to higher community awareness. However, it also states that “no studies have yet been conducted” to examine this link. This creates a tension between implied causality and admitted uncertainty. To maintain scientific clarity, this should be reframed as a hypothesis generated from qualitative insights, rather than an established effect. For example, the authors might say: “Participants perceived greater IPC adherence at the TB hospital, potentially linked to community awareness. This observation warrants further empirical evaluation to determine its validity and generalizability.”

In essence, this manuscript addresses a critically important area for public health in low-resource settings. It offers thoughtful reflections on system-level IPC challenges and brings forward actionable findings. If the aforementioned revisions are made, I believe the manuscript will make a meaningful contribution to IPC policy discourse in similar health systems.

Reviewer #2: Thank you for the opportunity to review this paper. The topic is highly relevant and critical, particularly for improving infection control practices in LMIC settings like Bangladesh. The article presents a clear methodological structure and includes rich contextual detail. I commend the team’s use of grounded theory and methodological triangulation. Below are my specific comments for clarity, accuracy, and consistency.

1. The first sentence of the introduction is a strong opening, but could benefit from one or two relevant citations to back up the claim beyond general statements.

2. “Among the total postoperative patients, an alarming rate of 41.7% developed a surgical site infection [4].” ⟶ Please consider whether “alarming” is too strong for a scientific tone. A more neutral phrasing could be “a high rate of…”

3. “This emphasises why preventing HAIs by maintaining IPC measures is crucial…” ⟶ This sentence would benefit from specifying how IPC contributes to economic and health system stability.

4. “For this study, grounded theory with an inductive approach was adopted…”⟶ You may briefly clarify what grounded theory offers for this topic—i.e., why it is suitable for studying system-level factors.

5. “Ethical permission was taken from the Ethical Review Committee, Bangladesh University of Health Sciences…” ⟶ Please mention the ethical approval number in parentheses.

6. The constant comparative method was adopted…” ⟶ Please add a citation here for readers unfamiliar with the method.

7. “Two investigators independently reviewed the primary codes to form axial coding…” ⟶ Please add how disagreements were resolved (e.g., consensus, third reviewer).

8. The codes were used to identify patterns among them and generate categories and, eventually, themes.” ⟶ Please clarify the analytic flow (e.g., from open codes → axial categories → selective themes) for transparency.

9. “For data management and analysis, the software Tageutte was used.”⟶ Please check spelling: did you mean “Taguette”? If so, you might also cite the software formally.

10. “In the KIIs, the minimum age and work experience in their role were 28 years and 7 months, respectively (Table 2).” – Please report a range or mean ± SD instead of only the minimum values.

11. If possible please include a summary table of key themes, sybthemes, and representative quotes to improve the readability.

12. “Most participants admitted the presence of HAIs… but were unable to provide reliable or specific data.” – This is a critical insight. Consider linking this finding to the absence of a routine data audit. This helps frame it as a structural failure rather than an anecdotal gap.

13. “The training isn’t regular and is usually reserved for nurses in senior positions.” – This quote is powerful and actionable. Please consider distinguishing between knowledge gaps due to poor cascade training vs. a complete lack of training infrastructure in the discussion.

14. Theme 4 and theme 6 shows conceptual overlap. Please consider combining the themes into a larger theme.

15. For the theme 10, recommendations, these are quite long and anecdotal. Please consider regrouping in policy-level, facility-level or community engagement strategies.. etc.

16. “HAIs impact the health of both healthcare workers and patients…which is why the prevention of HAIs is key…”- This is kind of repetition of general knowledge. Please reframe it to avoid redundancy.

17. “According to this study’s findings, currently, there is no accurate database…”– This is important but needs clearer linkage to national policy implications.

18. “Besides uncheck HAI rate and gaps in IPC knowledge…”– Minor typo: should be “unchecked HAI rate.” Also, I suggest more formally framing this as “unmonitored HAI burden.”

19. The sentence “However, no studies have yet been conducted to understand the role and impact of community awareness in maintaining IPC measures…”– This is a strong knowledge gap; please highlight it earlier and frame it as a future research priority. Consider referencing any literature on community-based IPC if available.

20. The section about social obligations and visitors – This is contextually rich. However, consider distinguishing between cultural determinants and structural health system failures; both deserve separate policy solutions.

21. Triage and zoning – Excellent recommendations. However, I suggest aligning them with WHO IPC guidance or WASH in health care facilities guidelines.

.

Reviewer #1: **Yes:**Dr. Awsaf KarimDr. Awsaf KarimDr. Awsaf KarimDr. Awsaf Karim

Reviewer #2: No

---

## [Author Response · Author response to Decision Letter 1]

12 Jan 2026

We thank the Editor and the Reviewers for their careful evaluation and constructive comments. We have revised the manuscript accordingly. A detailed, point-by-point response to all reviewer and editor comments is provided in the uploaded document entitled “Response to Reviewers”, in which each reviewer comment is reproduced and followed by our response and the corresponding changes made in the manuscript. Major revisions include clarification of the methodology, and discussion, as detailed in the response document.

---

## [Decision Letter · Decision Letter 1]

16 Feb 2026

Dear Dr. Sultana,

Thank you for submitting your manuscript to PLOS ONE. After careful consideration, we feel that it has merit but does not fully meet PLOS ONE’s publication criteria as it currently stands. Therefore, we invite you to submit a revised version of the manuscript that addresses the points raised during the review process.

We look forward to receiving your revised manuscript.

Kind regards,

Helen Howard

Staff Editor

PLOS One

Journal Requirements:

Reviewers' comments:

Reviewer's Responses to Questions

**Comments to the Author**

Reviewer #1: All comments have been addressed

Reviewer #2: (No Response)

2. Is the manuscript technically sound, and do the data support the conclusions?

Reviewer #1: Yes

Reviewer #2: Yes

3. Has the statistical analysis been performed appropriately and rigorously?

Reviewer #1: Yes

Reviewer #2: N/A

4. Have the authors made all data underlying the findings in their manuscript fully available?

Reviewer #1: Yes

Reviewer #2: Yes

5. Is the manuscript presented in an intelligible fashion and written in standard English?

Reviewer #1: Yes

Reviewer #2: Yes

Reviewer #1: Thank you for submitting the revised version of your manuscript. I have reviewed the responses to my comments and the corresponding revisions, and I find that they have been addressed adequately. I appreciate the clarity of the revisions and the care taken to respond to the points raised. I have no further comments at this stage. I wish the authors all the best for their publication.

Reviewer #2: I appreciate the authors’ thoughtful revision in response to my comments. The expanded justification for the use of grounded theory and the added emphasis on system-level processes, interactions, and organizational dynamics strengthen the rationale and demonstrate careful consideration of the earlier comment.

However, the justification remains somewhat abstract and would benefit from greater methodological specificity. In particular, it would be helpful for the authors to clarify how grounded theory was operationalized in this study and what it uniquely enabled beyond a descriptive qualitative approach (e.g., thematic analysis) in this study.

.

Reviewer #1: **Yes:**Dr. Awsaf KarimDr. Awsaf KarimDr. Awsaf KarimDr. Awsaf Karim

Reviewer #2: No

---

## [Author Response · Author response to Decision Letter 2]

25 Feb 2026

Thank you for this valuable suggestion. We have revised the method section to clarify how grounded theory was operationalized (Page 4, lines 77-80) and the discussion section to clarify how this approach enabled the development of a contextual explanation of system-level drivers of hospital-acquired infections beyond a descriptive thematic account. (Page 19, lines 443-448)

---

## [Editor Report · Decision Letter 2]

15 Mar 2026

Investigation to Mitigate System-Level Factors Contributing to Hospital-Acquired Infection Cases in the Low-Resource Setting: A Qualitative Exploratory Study in Bangladesh

PONE-D-25-52579R2

Dear Dr. Sultana,

We’re pleased to inform you that your manuscript has been judged scientifically suitable for publication and will be formally accepted for publication once it meets all outstanding technical requirements.

Kind regards,

Marianne Clemence

Staff Editor

PLOS One
---

## [Editor Report · Acceptance letter]

PONE-D-25-52579R2

PLOS One

Dear Dr. Sultana,

I'm pleased to inform you that your manuscript has been deemed suitable for publication in PLOS One. Congratulations! Your manuscript is now being handed over to our production team.

Kind regards,

on behalf of

Dr Marianne Clemence

Staff Editor

PLOS One